# Risks Without Borders: A Cultural Consensus Model of Risks to Sustainability in Rapidly Changing Social–Ecological Systems

**Berill Blair [1,*] and Amy L. Lovecraft [2]**

[1]  Environmental Policy Group, Social Sciences Department, Wageningen University and Research, Hollandseweg 1, 6706KN Wageningen, The Netherlands

[2]  Political Science Department, University of Alaska Fairbanks, PO Box 756420, Fairbanks, AK 99775-6420, USA; allovecraft@alaska.edu

*  Correspondence: berill.blair@wur.nl

**Abstract:** Global sustainability goals cannot realistically be achieved without strategies that build on multiscale definitions of risks to wellbeing. Particularly in geographic contexts experiencing rapid and complex social and environmental changes, there is a growing need to empower communities to realize self-identified adaptation goals that address self-identified risks. Meeting this demand requires tools that can help assess shared understandings about the needs for, and barriers to, positive change. This study explores consensus about risks and uncertainties in adjacent boroughs grappling with rapid social–ecological transformations in northern Alaska. The Northwest Arctic and North Slope boroughs, like the rest of the Arctic, are coping with a climate that is warming twice as fast as in other regions. The boroughs are predominantly inhabited by Iñupiat people, for whom the region is ancestral grounds, whose livelihoods are still supported by subsistence activities, and whose traditional tribal governance has been weakened through multiple levels of governing bodies and institutions. Drawing on extensive workshop discussions and survey experiments conducted with residents of the two boroughs, we developed a model of the northern Alaska region's social–ecological system and its drivers of change. Using cultural consensus analysis, we gauged the extent of consensus across the boroughs about what key risks threaten the sustainability of their communities. Though both boroughs occupy vast swaths of land, each with their own resource, leadership, and management challenges, we found strong consensus around how risks that impact the sustainability of communities are evaluated and prioritized. Our results further confirmed that rapid and complex changes are creating high levels of uncertainties for community planners in both boroughs. We discuss the mobilizing potential of risk consensus toward collective adaptation action in the civic process of policy making. We note the contribution of cultural consensus analysis as a tool for cross-scale learning in areas coping with rapid environmental changes and complex social challenges.

**Keywords:** risks; sustainability; cultural consensus; social–ecological system; Arctic

## 1. Introduction

The cumulative effects of resource extraction [1] and climate change [2] have contributed to an increasingly challenging decision environment in Northern Alaska. Communities who depend heavily on the availability of subsistence resources for their livelihoods are especially pressured by climate change impacts as well as rapid social, political, and economic drivers of change [3]. Ramping, large-scale shifts in ecosystems, such as thawing sea ice and permafrost, coastal erosion, weather dynamics, and animal migration patterns, can cause deterioration in food supply and vital infrastructure [4]. These challenges are exacerbated by political, social, and economic tensions stemming

from simultaneously managing viable, sustainable resource development and optimal environmental protection that is in accordance with tribal, municipal, regional, state, and federal interests. Current institutional provisions at times fail to equitably include the views of Alaska's Indigenous stakeholders on risks, opportunities, and acceptable trade-offs between the two [5]. This phenomenon is not unique to Alaska, and human development dialogues in the wider Arctic context have placed emphasis on fate control as a measure of community capacity to steer developments toward desired and sustainable futures [6]. Fate control refers to one's ability to guide their own destiny, reflected in political and decision-making power, economic control, access to knowledge, and recognition of human rights [7]. In other words, the sustainability of Arctic Indigenous communities is affected by the extent to which they are free to self-govern within a complex political landscape that demands multi-level negotiations of development planning. Tensions from contrasting perspectives about appropriate action across levels of government inhibit self-governance and fate control and negatively affect adaptation outcomes. On the other hand, when institutions, such as those that manage risks, fit the social as well as the biophysical domains within which they operate, they can be effective in supporting sustainable futures [8]. As individuals, communities, and entire social–ecological systems are adjusting to a changing climate, bottom-up perspectives about risks must inform top-down approaches to adaptation [9]. The negotiation of these multi-level perspectives bears great impact on the fate control of Arctic communities.

Because local control is important for Arctic community sustainability and because effective institutions must accommodate the social and environmental scales that they oversee, optimal adaptation outcomes demand productive collaboration and the scaling of shared visions across jurisdictions. However, there is a continuous push-and-pull between lower and higher institutional levels, and the extent to which each may make decisions autonomously or interdependently with other levels. This problem of scale is complex, and top-down devolution of control alone is not the solution [10]. Empowerment across institutional levels demands bottom-up approaches as well, such as the aggregation of reciprocal interests at lower levels to create larger jurisdictional reach [10,11]. This study is an exploration of the existence of a shared risk subsystem [5] among lateral or same-level entities in the political hierarchy in the northern Alaska region. We explore the extent to which risks to sustainability may scale across two same-level jurisdictions, the North Slope Borough (NSB) and the Northwest Arctic Borough (NAB), and frame our findings to discuss potential implications for collective, adaptive action.

## 1.1. Research Objectives and Questions

In this study, we focus on issue consensus among experts from the two boroughs by analyzing shared beliefs about community sustainability and risks. The two boroughs each occupy a vast geographic region (approximately 135,545 mi$^2$/218,138 km$^2$) larger than the country of Italy, and though there are similarities in social–ecological systems and cultural heritage, they represent unique jurisdictional entities in Alaska's political hierarchy and differ greatly in available resources, local and regional concerns, and external pressures. Our objective is to explore whether a shared, regional vision of community sustainability and agreement about risk priorities can be identified.

We consider the following questions and methods of analysis:

(1)　To what extent do the boroughs share similar views about the speed and complexity of changes that impact their social–ecological systems?

- We rely on workshop results about risks and a cross-impact analysis of uncertainties in our stakeholders' decision environment [12] to answer this question.

(2)　To what extent do the two boroughs have shared beliefs about which risks threaten the sustainability of their communities the most?

- To identify shared perceptions, we look for a cultural consensus model using cultural consensus analysis [13,14].

The first question probes whether the two boroughs perceive similar levels of uncertainty posed by the rapid and complex changes occurring in their communities. The greater the uncertainty in the decision environment, the more difficult it becomes to see how current plans and actions shape future outcomes [15]. The second question aims to gauge whether the two boroughs have a shared vision about which risks should have priority on the political agenda. The identification of reciprocal needs can help to aggregate interests across jurisdictions and promote collective action [10,11]. On the contrary, significant differences in perspectives decrease cross-scale collaboration, which, in turn, decreases available capital to respond to emerging risks [16].

## 1.2. Conceptual Framework

The analytical framework in this paper relies on the resilience and adaptation literature, building on theories of adaptive capacity and vulnerability in complex social–ecological systems [17–19]. We chose the resilience and adaptation theory framework because it is especially well suited for the study of decision making across scales in linked social–ecological systems during times of rapid change [20–24]. Furthermore, resilience and adaptation literature acknowledges that collaboratively defined problem definitions are important for encouraging adaptation across levels of governance [19] and that successful adaptation, in turn, can help to sustain livelihoods [25]. These are well-suited analytical perspectives in the context of rural Arctic communities who engage in subsistence activities and their capacity to adapt to rapid social and environmental change. For the purposes of this study, the vulnerability and adaptive capacity of social–ecological systems will be considered in terms of the communities' ability to achieve outcomes that are compatible with their desired futures.

The concept of social–ecological systems is central to resilience literature. From the original framework, intended to analyze the resilience of mutually dependent systems of people, institutions, and nature [26], to later models, focusing on robustness and non-linear relationships [27], the concept has been extensively used in social and environmental sciences [28]. At their core, social–ecological systems help to understand the complex and often non-linear interactions between social and ecological processes. In resilience literature in particular, a social–ecological system lens seeks to respond to and shape a rapidly changing world [18]. Vulnerability is another essential analytical lens in the resilience and adaptation literatures. Vulnerability is the degree to which a system is likely to experience harm due to exposure to hazards [25,29], given the system's adaptive capacity or its ability to respond (avoid, eliminate, or mitigate) [18]. For example, a community located in proximity to geophysical processes that produce earthquake events is only vulnerable to the extent that their capacity to plan and implement mitigative measures to respond to shock events is insufficient for preventing loss and damage. In other words, the degree to which a community is likely to experience harm due to their exposure to a hazard is a combination of internal factors, such as sensitivity and adaptive capacity, and physical or external factors due to exposures [30].

Adaptive capacity is the ability to adjust to changing conditions to achieve resilience, the maintenance of desirable systems states or outcomes [31–33]. Adaptive capacity is relative and dynamic, and is supported by resources based in social, human, and natural capital [34]. In efforts to reduce vulnerability, the increasing of adaptive capacity across multiple levels in a social–ecological system is an important tool [18,22,30]. The implementation of adaptation approaches requires social capital [31]. Social capital refers to having the agency needed to act in one's own interest; for example, having fate control. It can also include social learning by building consensus, empowering stakeholders to adapt, reducing conflicts, and increasing fairness [35]. However, at both the individual and institutional (policy) levels, adaptation hinges on whether impacts, processes, and events are perceived as risks that require a response [36]. Social and cultural values drive risk perception, and, in turn, the extent to which it becomes a limit to adaptation [36]. Group-level perceptions can take the form of

shared knowledge and attitudes that, when strong enough, can be considered a shared cultural model, such as a cultural model of risks, as a normative set of beliefs [13,14].

The rules and regulations, such as the institutions that govern risks, can hinder or promote adaptive action. Vulnerability, social capital such as fate control, and adaptive capacity are all influenced by the effectiveness of institutions that facilitate adaptive action [8,37]. Research has shown, for example, that sovereignty and effective institutions are prerequisites for successful economic development on tribal lands [38]. Effective institutions, in turn, are a function of two things: Cultural legitimacy, or the perception that institutional norms and objectives fit cultural norms, and 'sovereignty in practice,' which refers not to any legal definition of sovereignty, but to the extent to which sovereignty is allowed to be exercised in practice [38]. Sovereignty in practice and fate control are therefore kindred concepts, as they are both concerned with the degree of self-determination flowing through institutions, practices, and relationships. Other lines of research [39] link sovereignty in practice to the institutional development literature to highlight the importance of local units of collective action as building blocks in higher-level political organization [40–42].

From these links among sovereignty in practice, cultural legitimacy, and local units of collective action, two points emerge as relevant to our study. Firstly, sovereignty, fate control, and adaptation move through or are impacted by the institutions that govern the distribution of resources and power. The institutions that establish the extent to which groups of people, such as those of Alaska's tribal villages, are allowed to exercise fate control and make decisions about adaptive action become instrumental in sustainability outcomes. Secondly, organized local units can navigate these corridors of power better than isolated ones. Shared uncertainties and risks allow resources and political will to be pooled across jurisdictional boundaries. Such aggregation of interests can increase jurisdictional reach if collective action is taken to bring attention to shared concerns at higher levels of governance. Cross-jurisdictional collaboration can therefore help reduce institutional limitations to adaptive action.

### 1.3. Study Area

This study was conducted in and with communities of the northern Alaska region (Figure 1). Many of the risk issues discussed herein, and the social and ecological drivers of change that create the complex challenges for sustainability governance, are not unique to this region. Coastal communities around the world may well identify, for example, with the planning and development challenges surrounding flood hazards, which are projected to increase in coming decades [43]. However, the Arctic is warming twice as fast as other regions [44]. Because a changing Arctic cryosphere has a far-reaching footprint of influence in distant geographies, constructive multilevel adaptive actions are needed [45]. Northern Alaska's communities, in this sense, are laboratories for learning, through which a global audience can glimpse their own future challenges related to collective adaptive action.

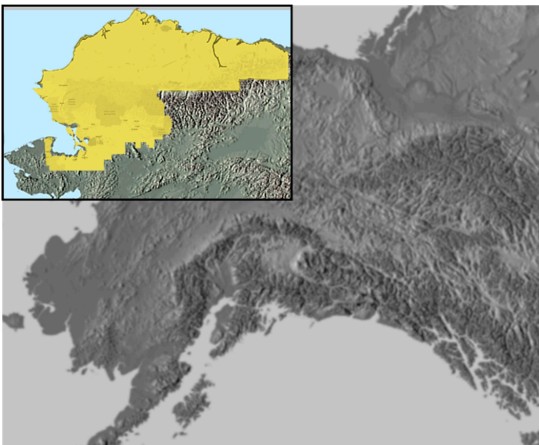

**Figure 1.** Study area. The yellow highlighted area encompasses the North Slope and Northwest Arctic boroughs (State of Alaska, USA).

The northern Alaska region, Arctic Alaska, is comprised of the North Slope (NSB) and Northwest Arctic (NAB) boroughs. The two boroughs together occupy approximately 135 thousand square miles, inhabited by a total of 17 thousand people. Of this land mass, the NSB is a 95 thousand square mile area with approximately 9800 residents. The NAB has about 7000 residents across a 40 thousand square mile area. In both boroughs, the population is about 80% Iñupiat Alaska Natives, but this number can be as high as 90%–95% outside the hub cities in small villages [46,47]. The Iñupiat have a long history in this region, going back thousands of years. Core Iñupiaq cultural values and Traditional Knowledge systems are still very important, and this is true for all Alaska Native cultures across Alaska [48]. For the Iñupiat people, subsistence is still a way of life and an essential component of livelihood in rural Alaska [49].

The NSB is bordered by the foothills of the Brooks Range to the south and the Beaufort Sea to the north. The NSB was incorporated in 1972 largely to maximize local determination through centralized, regional representation at state and federal levels and the right to tax oil companies [50]. The NAB has its coastline on the Chukchi Sea. The NAB followed suit and incorporated in 1986. The economy of the NAB relies heavily on the Red Dog Mine, the world's largest zinc and lead mine. The NAB levies no taxes on the mine, but relies on negotiated payments in lieu of taxes for funding of basic services and village improvements. The NSB collects property tax revenues from oil facilities. In both boroughs, industrial activities are viewed as a double-edged sword. On the one hand, they help to finance essential services, but on the other hand, communities have serious concerns about the environmental and health effects of pollution [51,52].

In 1968, the discovery of oil in Prudhoe Bay on the coastline of what is NSB today set into motion a powerful chain of events, and ultimately culminated in the settlement of land claims for Alaska Native tribes in 1971. The Alaska Native Claims Settlement Act endorsed land titles to 44 million acres of land and approximately 1 billion USD, to be managed by twelve for-profit Alaska Native regional corporations. These twelve regional corporations were also granted subsurface land rights. The Act converted communal, aboriginal claims of Alaska Native tribes into private property rights through shares of stock in over 200 various Native regional, village, and group corporations [53]. Both boroughs have a regional for-profit and non-profit corporation. The NSB has eight and the NAB has eleven villages who are Alaska Native tribal entities and have their own tribal governments.

If we take the example of a resident of Utqiaġvik, the seat of the NSB, they are subject to rules and regulations at the borough, state, and federal level in the political hierarchy. At the same time, they are also represented by The Native Village of Barrow Iñupiat Traditional Government, which is the tribal (village) council, the Iñupiat Community of the Arctic Slope, which is the regional tribal entity, and the Alaska Federation of Natives, the state-wide Native organization. Additionally, they have shares in the Ukpeaġvik Iñupiat Corporation, which is the village for-profit corporation, and in the Arctic Slope Regional Corporation, which is the regional Native corporation. Health and social services are administered by the Arctic Slope Native Association as the region's non-profit Native corporation. These entities form a complex web of traditional and modern governance structures and institutions, each with their unique role in community adaptive capacity. Though Alaska's 229 tribes are legally sovereign entities, with jurisdiction over their domestic affairs, their councils' authorities are limited to powers over people (via membership), but not over place or territory, which is in the hands of corporate leadership [54]. The idea of land ownership itself as now exists through corporate governance is quite different from the traditional relationship to homelands. As an Elder living in an NSB village put it: "We belong to the land. The land does not belong to us" [55]. Nevertheless, local control over what happens in homelands is very important to community sustainability in the region [56].

Both boroughs have experienced profound impacts from rapid climate change, due to warming temperatures, changing precipitation patterns, sea ice loss, coastal erosion, river flooding, ecosystem changes, and permafrost thaw [57]. These changes have caused shifts in the availability of subsistence resources and exerted immense pressures to take adaptive measures in the region's communities. During our visits with local residents in the region, the impacts on everyday life from environmental

changes were an important discussion point: "Less snow makes it hard to travel and increases distance and time [for subsistence activities]" shared one resident [58]. "Permafrost is one or two feet lower than a few decades ago, and our underground cellars are at risk" shared another [59], though residents also emphasized that their communities have a long history of adaptability. However, the rate and extent of bio- and geophysical changes are surpassing historic patterns and are increasingly testing the limits of adaptive capacity in these communities. For example, The Native Village of Kivalina, the northern-most community in the NAB, is currently under pressure to relocate away from their current coastal location due to rising sea levels and coastal erosion. Kivalina is one of several communities in Alaska facing imminent relocation [60].

## 2. Methods

Prior to recruitment and data collection, the research procedures were approved for use with human subjects (University of Alaska Fairbanks IRB# 496953-4), and informed consent was obtained from all participants.

### 2.1. Participants

A series of three workshops provided our platform for this research [49]. Residents from both boroughs participated in these workshops in 2015–2016 to discuss the future of healthy, sustainable communities in the region. We invited participants with expertise across different sectors, including subsistence, Iñupiaq culture, environment, education, health, government, economy, emergency and disaster response, public safety, and justice. Two workshops were hosted in the headquarters of the two boroughs: Utqiaġvik (formerly known as Barrow) in the NSB (total participants: N = 29; breakdown by participants' borough of residence: $n_{NSB}$ = 19, $n_{NAB}$ = 10), in Kotzebue in the NAB (total participants: N = 24; breakdown by participants' borough of residence: $n_{NSB}$ = 9, $n_{NAB}$ = 15), and one workshop on neutral grounds, so to speak, in Anchorage in southcentral Alaska (total participants: N = 18; breakdown by participants' borough of residence: $n_{NSB}$ = 10, $n_{NAB}$ = 8). We recruited participants by taking part in community meetings, talking to individuals directly, making personal phone calls, and sending formal email invitations. The recruitment targeted key community figures, who are residents in one of the two boroughs and who have expertise in at least one of topics mentioned above. In total, 51 participants attended at least one of our workshops, and we had 17 return participants who attended at least two.

### 2.2. Survey Instrument

To capture views about perceived uncertainties due to the ongoing changes in the social–ecological system, the survey used a dedicated set of questions in a cross-impact analysis format. This questionnaire asked participants to reflect about the speed and complexity of changes that are impacting the northern Alaska region (Supplementary Material S1). High scores on both the complexity and the speed of changes indicate a difficult planning environment and greater uncertainty predicting future outcomes [12]. The evaluations used a seven-point Likert scale format about perceptions of complexity (13 evaluative statements) and speed of changes (12 evaluative statements). Charted along two axes, low scores indicate continuity in, and foresight of, the system's future states. Higher scores indicate a shift toward uncertainty and a difficult planning environment.

To identify shared attitudes and perceptions about risks to community sustainability in the two boroughs, we used the cultural consensus analysis (CCA) method [13,14]. CCA estimates shared beliefs relying on three basic steps [61]. First, it finds shared knowledge using principal component analysis; secondly, it tests individual competence of that model by assigning so-called competence scores; finally, it aggregates individual answers to questions by weighting the final cultural model in favor of respondents with high competence. In other words, in CCA, the influence of informant guessing is minimized, enabling the researcher to describe cultural spaces inhabited by respondents. It is very important to note that culture and cultural competence are used in very context-specific

ways in CCA. Culture here refers to the shared sets of beliefs among a specific group of people; in our case, the select group of NAB and NSB residents who participated in our workshops. What CCA terms "competence" is the individual's level of knowledge of the knowledge domain held true by this particular group of people and about the particular line of inquiry presented herein. The authors were hesitant to employ the term "competence score" in this study to avoid confusion. In the end, we keep with the CCA tradition of using this terminology, but caution the reader that any reference to cultural competence in this study does not in any way represent an individual's competence in other knowledge cultures, such as in their Iñupiaq culture or Traditional Knowledge, for example.

At the first workshop in Utqiaġvik, we asked participants about risks to community sustainability through an open-ended survey question. The written results from this question helped to establish a cultural domain of risk, an important step for CCA, to ensure culturally relevant question items later on. Responses were analyzed using a grounded theory approach [62] and iteratively coded in ATLAS.ti (v.8.0). This textual analysis produced a total of 187 observations about risks, which were sorted using textual content into distinct risk themes. The final 40 codes in the 7 dominant risk themes that emerged were: Decision making, health and health care, environmental change, cultural changes, industrial activities, education, and cost of living (Supplementary Table S2). This list of mainly socioeconomic risks was supplemented from literature about land cover changes in Northern Alaska [63] to probe perceptions of risk related to the natural environment as well. This resulted in 7 additional environmental change items for CCA, presented to participants in subsequent workshops.

Overall, fourteen potential risk items were analyzed by 28 participants at the second and third workshops. According to CCA best practices [14], two participants were excluded from analysis due to abandoned, incomplete questionnaires. This left 26 valid data sets ($n_{NSB}$ = 12, $n_{NAB}$ = 14). The match coefficient method of the formal consensus model was used [13] in the UCINET software package [64].

## 3. RESULTS

### 3.1. Northern Alaska Social–Ecological System: Speed and Complexity of Changes

During the analysis of written responses to the open-ended risk question "what are the top five issues that threaten the success of healthy, sustainable communities in 2040?", we checked co-occurring codes and relationships using the network view manager of ATLAS.ti [65]. The resulting heuristic model of causal and associative relationships among social–ecological system components was complemented by notes taken at the workshop during roundtable and plenary discussions. This final model was depicted in a stylized illustration of drivers of change in the northern Alaska region, as understood by participants (Figure 2).

Through this model, a picture of a complex social–ecological system emerged with numerous, interconnected drivers of change. Respondents frequently attributed observed environmental changes to climate change, and linked its impacts to infrastructure degradation and changes in subsistence resource availability. Subsistence activities are vital to food security, but climate change impacts combined with development increasingly impact access, according to participants. Diminishing sea and river ice decrease hunter mobility, while the combination of reduced sea ice and industrial activities increases marine traffic. The reduced extent of sea ice is responsible for coastal erosion, which threatens an already fragile infrastructure.

An overarching theme implicated decision making, mostly at the state and federal levels, in excluding local interests from important decisions. Local political processes were indicated to a lesser degree in hindering healthy, sustainable communities. The theme *divided local interests* represents survey responses such as "disagreements among local governments", "lack of unity in terms of what is most important", and "lack of support and collaboration locally." As one participant remarked: "We are all Iñupiaq, not corporation against corporation."

According to respondents, insufficient authority over decisions that impact local social–ecological conditions affects community adaptive capacity. Respondents described a struggle to "walk in both

worlds", referring to the simultaneous participation in subsistence and wage employment activities. This can occur due to scheduling, lack of time to learn, high costs, displacement by industry, and the education system failing to prepare youth for either world. Participants reported that those who leave for education often do not return to their communities. Others move away for jobs, or, as one participant observed, "children are educated to leave" if there is inadequate access to vocational training and good-quality, culturally appropriate education that would allow local youth to learn Traditional Knowledge.

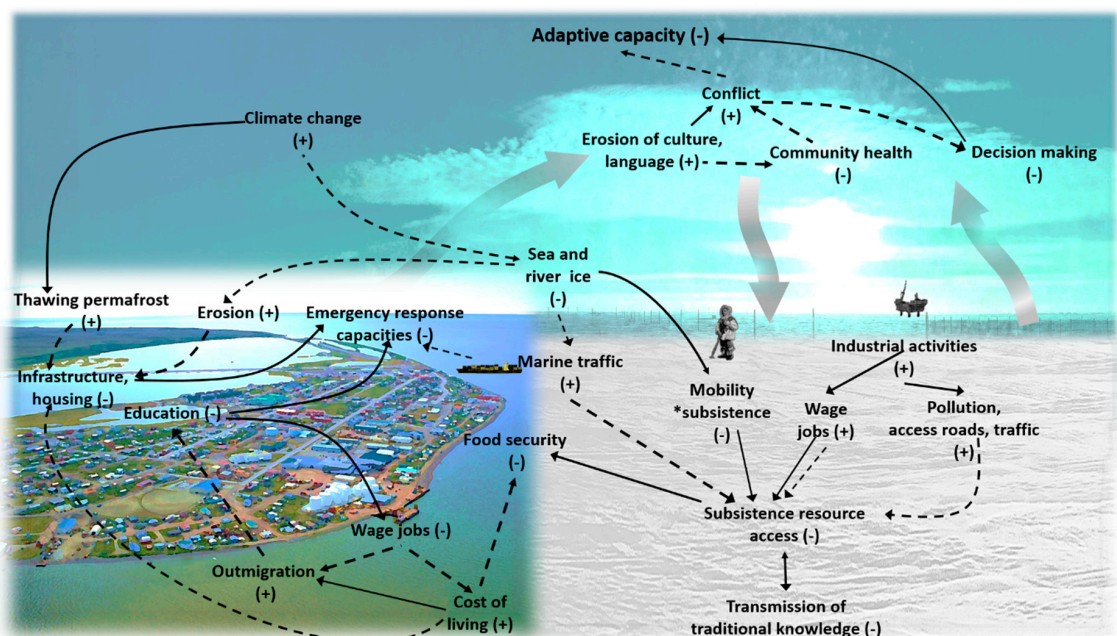

**Figure 2.** Model of social–ecological drivers of change and community impacts. The model was described by workshop participants (N = 47). Solid lines represent reinforcing or positive relationships, dashed lines represent weakening or inhibiting relationships between system components. Pluses indicate increasing trends, minuses indicate decreasing (or deteriorating) trends. The model's design concept draws on Hopping et al. (2016) [66].

Wage employment is scarce, and, while it provides cash funds to assist with the costs of subsistence activities, it reportedly also takes time away from them. The high cost of living encourages outmigration, which impacts the availability of teachers and funding for education. Respondents admit that industrial development often provides jobs, but pointed out that it also affects subsistence resources because of noise and air pollution, altering animal migration patterns, and causing hunters to travel longer distances to find game. One respondent pointed out that the combination of increased marine traffic, inadequate infrastructure, and the few local, trained personnel decreased emergency response capacities. All of these drivers of change affect culture (language, heritage) and transmission of Traditional Knowledge. As culture changes, socioeconomic pressures and conflict increase and community health declines, according to participants.

Informants from both boroughs described aspects of these same processes. Slight differences were noted in terms of frequency of emergence or priorities among these factors. The most frequently mentioned risk, ineffective decision making, was the same in both boroughs. The second most frequent were health and health care issues. In third place were industrial activities tied with environmental changes in the NSB group, while cost of living and health and health care issues came in second and third (respectively) in the NAB group. This analysis of observed drivers of change proved significant in providing a generalizable picture of workshop participants' perceptions about a complex and rapidly changing northern Alaska social–ecological system.

Complementary to the description above, Figure 3 depicts results from the cross-impact questionnaire about uncertainties in the region's social–ecological system. The figure relies on a coordinate system to reflect median values of the two groups, situated along the dimensions of speed and complexity of changes. Situating the level and type of uncertainty according to the chart helps us to understand the extent to which participants feel that they can govern their region toward desired future states. Both groups scored median values around 5 on both the speed of changes and complexity (NSB speed of changes = 5, complexity = 4.5; NAB speed of changes = 4.75, complexity = 5). This indicates that our participants perceived their social–ecological system to be impacted by rapid, complex changes, toward the higher end of the evaluative scale. These shared, representative perceptions about drivers of change, risks, and uncertainties have relevance for finding common ground as building blocks for shared strategies in planning sustainable communities.

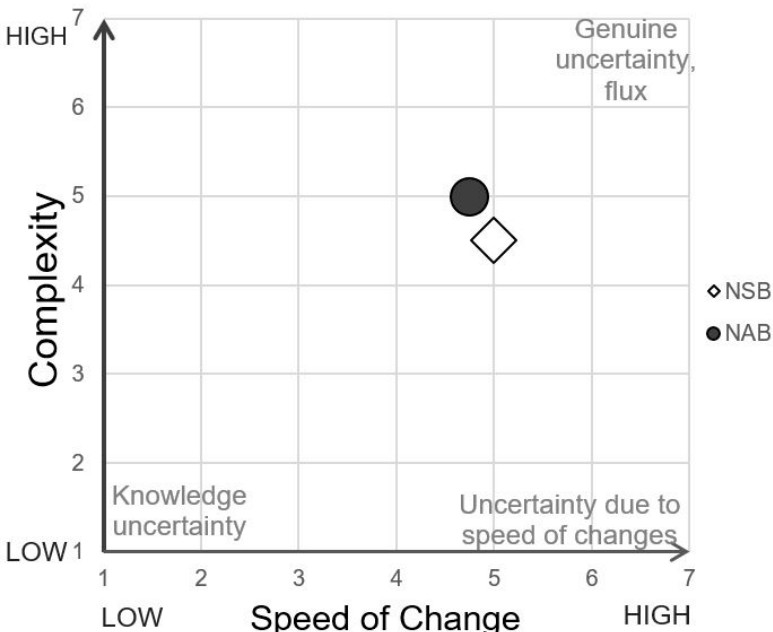

**Figure 3.** Uncertainty as a factor of the speed and complexity of changes in northern Alaska's social and environmental systems, as perceived by North Slope Borough (NSB) and Northwest Arctic Borough (NAB) participants. The chart indicates types of uncertainty as factors of complexity and change levels, adapted from Lindgren and Bandhold (2009) [12].

*3.2. A Cultural Consensus View of Risks*

Fourteen risk items were subjected to CCA to explore the extent to which risks that threaten community sustainability are shared between the boroughs. In CCA, the results are a good fit to the consensus model if there is a presence of a single factor that explains most of the variation in the responses, with a first-to-second eigenvalue ratio greater than or equal to 3.0. The CCA results indicated a good fit to the consensus model with a first-largest eigenvalue (18.8) to second-largest eigenvalue (1.5) ratio of 12.35, indicating a one-culture domain among the informants. The overall group cultural competence score was high at 0.8, SD = 0.3 (NSB cohort = 0.7, SD = 0.4, NAB cohort = 0.9, SD = 0.1), indicating that as a group, participants on average gave 80% correct answers to the shared cultural domain. There was one negative competence score (<5% of respondents). A negative competence score can occur when the method's function to correct for guessing pushes the lower limit of adjusted matches from 0 toward −1 [13]. A negative score signals that a participant responded very differently from the dominant knowledge culture model. Still, the results convey a consensus model with otherwise very high competence scores, a good fit, and clustering of agreement. The risk items, the culturally correct answers, and the frequency of responses are presented in Table 1. The consensus

analysis revealed that experts from the two boroughs have a basic agreement about the land cover changes and socioeconomic risks that threaten community sustainability in the region.

**Table 1.** Cultural consensus analysis results. The consensus-based answer column shows the culturally correct answer in line with the group's shared beliefs; the other columns show the number of correct replies overall and per borough.

| Risk Items Evaluated for CCA | CCA Final Results: Is Item Viewed as a Risk to Community Sustainability? | All N = 26 | NSB n = 12 | NAB n = 14 |
|---|---|---|---|---|
| Less snow in winter | Yes | 25 | 10 | 14 |
| Shallow river and lake waters | Yes | 24 | 10 | 14 |
| Thawing permafrost | Yes | 26 | 11 | 14 |
| More wildfires | Yes | 20 | 7 | 13 |
| Vegetation change | Yes | 22 | 9 | 12 |
| Later fall freeze-up | Yes | 25 | 12 | 12 |
| Earlier spring breakup | Yes | 25 | 11 | 14 |
| Health and health care issues | Yes | 23 | 9 | 13 |
| Environmental problems | Yes | 26 | 11 | 14 |
| Education issues: Formal schooling | Yes | 25 | 9 | 14 |
| Education issues: Transmission of traditional knowledge | Yes | 27 | 11 | 14 |
| Ineffective decision making | Yes | 25 | 10 | 13 |
| Industrial activities | Yes | 24 | 8 | 13 |
| Risks to culture | Yes | 25 | 10 | 15 |

Figure 4 visualizes the patterns of agreement via non-metric multidimensional scaling in UCINET [67]. The proportion of similarities indicated in the agreement matrix is represented here as a pattern of proximities in a multidimensional space (stress = 0.035, iterations = 50). The stress value is the distortion that occurs when data are transposed over multiple dimensions. Stress values below 0.1 are considered "excellent" or well representative of the patterns in the data [68], a criterion met by our model.

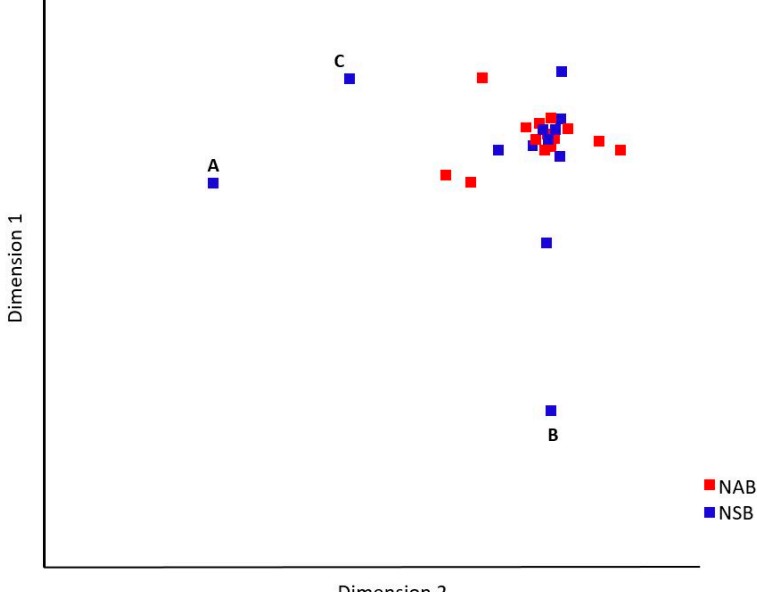

**Figure 4.** Nonmetric multidimensional scaling of agreement: NSB (blue) and NAB (red) respondents. "A" marks the respondent with negative competence score (−0.01).

The x and y axes do not represent meaningful numeric values in nonmetric multidimensional scaling beyond relative distance between objects. The axes anchor the points in space, providing coordinates for each object only to the extent that the between-node distances remain in proportion with the underlying (in this case, agreement) matrix. The consensus pattern can be seen clearly in the tight clustering and overlap of blue and red squares in the upper right of the plot. The cluster denotes respondents with high competence in the shared cultural domain with a combined competence score of 0.9 (SD = 0.1). Two informants had low competence scores (A, B) and occupy spaces somewhat distant from the cluster.

### 3.3. Summary Results

The strong consensus found for risk concepts confirmed that, although these boroughs operate in different parts of Arctic Alaska and work with different resources, contingents, pressure groups, and challenges, the overarching risks to sustainability are of a "one-culture model." The participants of this study were all experts in some area of community sustainability, and were educated both in the formal school system and in Traditional Knowledge. Their diverse backgrounds ensure that the shared cultural domain is inclusive of diverse sustainability perspectives. The one-culture risk model that has resulted from CCA is representative of this demographic. In other words, the results about risks to sustainability are likely scalable in the region, holding true for mixed groups of practitioners in education, subsistence, community health, criminal justice, government, and the region's youth and their perspectives on these issues.

We see that the participants' mental model of the region's social–ecological system (Figure 2) tells of mutually reinforcing feedbacks between social and environmental shifts and ever-increasing complexities and changes. These feedbacks and cascading effects can increase uncertainty and decrease adaptive capacity [29,32] because decision making under uncertainty inhibits the ability to anticipate and prepare for future risks and issues that may emerge [69–71]. Uncertainty tends to force decisions towards chaos, with great uncertainty about outcomes and no consensus on what to do, while known outcomes and consensus on how to handle them facilitate rational decision making. In northern Alaska, stakeholders identify high uncertainty in the social–ecological system, but there are issues over which they form consensus. Consensus over risks facilitates complex decision making, and keeps discourse away from chaos and uncontrollable outcomes (Figure 3). As a result, the scaling up of perceptions about exposure—such as the complexity and speed of changes for example, or vulnerabilities threatening wellbeing in the social–ecological system—becomes highly relevant in adaptation outcomes.

## 4. Discussion and Conclusions

Although Iñupiat communities have a substantial history of self-sufficiency, having exercised successful ecosystem stewardship and adaptability for millennia in their homelands, they must now navigate a complex, multiscale political hierarchy as they manage their communities and resources. Given this paradox, proposed strategies for sustainable management of the region's social–ecological system need to amplify localized needs and specialized tactics built on local expertise and Traditional Knowledge. Previous research has highlighted similar issues regarding adaptation in rural and Indigenous Arctic communities, pointing to the crucial role of self-identified adaptation goals and strategies to sustainability [72] and fate control in general [7]. However, this premise and our study are relevant beyond the Arctic context and in other geographic contexts grappling with uncertainties produced by climate change and socially constructed vulnerabilities. Especially in areas of rapid biophysical change and complex social problems, there is a growing need for empowering local actors in the civic process of policy-making at the local and regional scales [73].

Previous studies have shown that a CCA approach can be a valuable tool to assess shared cultural perceptions and understandings about needs, gaps, barriers, and potentials in building strategies for positive change [74,75], which can lead to opportunities to pool resources and expertise. Collective action, in turn, increases issue salience at higher political platforms. In northern Alaska,

the incorporation of boroughs is an example of self-determination through the carving out of local collective centers of power within the new political structure that displaced traditional tribal authority. Our results show that, regionally, there is a predominant consensus around risk priorities in northern Alaska communities, which surpasses administrative boundaries. This means that regional tactics to combat risks are possible and are not hampered by isolated risk cultures.

The significance of CCA as a tool and of the particular cultural domain uncovered in this chapter lie in the potential to facilitate lateral networks. Such networks can stimulate collective social learning and increase adaptive capacity [32]. Previous research suggests, for example, that shared perceptions about environmental change can facilitate communication, resource assessment, and human resource management in a manner that is appropriate within the context of a given community [76,77]. Agreement over sustainability issues, such as risks that threaten sustainable futures, makes possible the mobilization of resources for collective action. For many Indigenous communities specifically, such horizontal networks can enhance fate control via increased capacity to advocate for shared goals and a regional vision [39]. Regional visions, in turn, have an important role in communicating the regional appropriateness of adaptive responses and scaling up the potential for sustainable outcomes [9,78].

Global sustainability goals in times of extreme events, from climate change, growing energy demands, water and food insecurity, and biodiversity loss, demand multi-scalar collaboration. However, resource development goals often create high-stakes, low-consensus policy processes. The outcomes of this study underscore the utility of cross-scale learning using tools that facilitate culturally appropriate strategies and adaptive action. Risk issues that scale horizontally between same-level political systems can provide context for larger-scale learning, pooling of resources, and collective action at higher political levels. These can act as buffers in the social–ecological system against emerging risks and increase the capacity of social systems to take action. When systems situated at lower jurisdictional levels engage in such collaboration, they increase their political capacity to engage with dominant, higher level jurisdictions to advocate for salient local issues. In Arctic rural communities in particular, this increased capacity positively affects the sovereignty-in-practice of local governments in the management of their communities toward desirable futures. But in all regions confronting rapid changes and complex social issues, cross-scale learning can pave the way for the formation, and movement of, collective risk perceptions across and within scales and levels of governance, and enhance the plurality of knowledge that is fundamental for managing sustainability on a global scale.

**Supplementary Materials:** The following are available online at http://www.mdpi.com/2071-1050/12/6/2446/s1: Questionnaire S1: Cross-Impact Analysis Questionnaire, Table S2: Codebook: Textual analysis of top 5 risks. Open-ended question results.

**Author Contributions:** Conceptualization, B.B., A.L.L.; Data curation, B.B.; Formal analysis, B.B.; Funding acquisition, A.L.L.; Methodology, B.B., A.L.L.; Visualization, B.B.; Writing—original draft, B.B.; Writing—review & editing, B.B., A.L.L. All authors have read and agreed to the published version of the manuscript.

**Funding:** This research project was supported by the National Science Foundation's ArcSEES Program #1263850.

**Acknowledgments:** The authors are grateful to the workshop participants in the Northern Alaska Scenarios Project, who so generously contributed their time and expertise to this study. We also thank the reviewers who helped to improve on earlier versions of this manuscript.

**Conflicts of Interest:** The authors declare no conflict of interest.

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
