# Peer review of "Risks Without Borders: A Cultural Consensus Model of Risks to Sustainability in Rapidly Changing Social–Ecological Systems"

_sustainability, doi:10.3390/su12062446_

Round 1
Reviewer 1 Report
This is a nice paper and I see it's value. My comments mostly address the use of jargon and clarifying definitions. As an ecologist who dabbles in social-ecological systems, I see the interrelationship of these two systems to be at the root of SES. Your paper focuses on the response of the social system to rapid change in the environment and not the interplay of the two systems. I suggested that it is a social/environmental problem that lacks a 'system' angle. Or if it has a system component, it is social only. This might be esoteric but I think SES in its strick usage requires a system approach from both sides. That doesn't take away from your paper, it is an important insight into how communities respond to having threshold shifts in ecological services resulting in massive and relatively rapid changes in environmental conditions. I just want the usage language to be more specified. That all said, it is a nice paper after you address these minor edits and minor philosophic ponderings, it is ready to go! Nice work!
I always want to review these in word, but in the pdf, I had to attache comments in the document. I created a second document with the comments to the side and I hope that is clear.

Author Response
Please see the attached document containing the point-by-point response to reviewer. The file was renamed by the system after upload to author-coverletter-6527048.v1.docx and to Report Notes in another place.
In addition, we made the following changes:
- abstract has been revised
- slight title change
- Figure 3 is now depicting median values
We thank the reviewer for their suggestions and helpful remarks.

Reviewer 2 Report
The submitted manuscript defines and implements a methodology aimed at analyzing and assessing: i) the ongoing social and ecological changes; ii) the risks concerning sustainability of the local development processes perceived by the local communities. Perception of the ongoing social and ecological changes is studied through the methodology proposed by Lindgren and Bandhold, 2009. Building on this methodology, the risks related to sustainability of the local development processes were detected on the basis of the Cultural consensus analysis (CCA) approach. The two research questions reported above are addressed by means of workshops implemented as follows: “We recruited participants by taking part in meetings such as those of the Iñupiat Community of the North Slope, talking to individuals directly, making personal phone calls, and sending formal email invitations. The recruitment targeted key community figures, who are residents in one of the two boroughs and who have expertise in at least one of topics mentioned above. In total, 51 participants attended at least one of our workshops and we had 17 return participants who attended at least two.” (lines 226-231). The methodological approach is implemented with reference to two Alaskan contiguous areas: the North Slope Borough (NSB) and the Northwest Arctic Borough (NAB).
In my opinion, the study is very interesting and relevant as regards the issue of analyzing and identifying community visions, in order to build local development plans on the basis of public awareness and consensus, as regards ongoing socio-ecological changes and risk perception. However, the submitted manuscript needs substantial improvement before being accepted for publication.
In a revised version of the study, the authors should carefully address the following points.
i. Subsection 1.3 “Study Area.” As regards the NAB and NSB spatial contexts, a comparison of these areas and other international regional contexts should be implemented, in order to make the reader aware of the reasons which make the submitted manuscript interesting for the vast scientific and technical public of the readers of Sustainability.
ii. Section 4 “Discussion.” This section is poor. I would recommend the authors discuss their outcomes in the light of the available studies related to CCA, concerning local development planning processes, based on public awareness and consensus. I would recommend the authors analytically put in evidence the advancements implied by their study as compared to the current literature, in order to make the reader aware of the value added of the submitted manuscript.
iii. Section 5 “Conclusions.” I would recommend the authors discuss the exportability of their methodological approach to other national and international contexts different from NSB and NAB. This would imply the identification of the reasons for which the study implemented with reference to the two Alaskan boroughs is likely to be helpful in addressing similar issues related to other locations.
iv. The English is good. However, I would recommend the authors be careful to typos here and there. For example: “The written results from this question helped to established a cultural domain of risk, an important step to CCA, to ensure culturally relevant question items later on.” (lines 261-263).
Author Response
We wish to thank the reviewer for their contribution to our manuscript. In addition to the below revisions, we wanted to point out the following changes:
- abstract has been revised
- slight title change
- Figure 3 now depicts median values
Reviewer:
i. Subsection 1.3 “Study Area.” As regards the NAB and NSB spatial contexts, a comparison of these areas and other international regional contexts should be implemented, in order to make the reader aware of the reasons which make the submitted manuscript interesting for the vast scientific and technical public of the readers of Sustainability.
Author:
Please see a 1st paragraph added under section 1.3 that provides a rationale for local-global connections, and a broader scientific interest in the outcomes of this study beyond this region.
Reviewer:
ii. Section 4 “Discussion.” This section is poor. I would recommend the authors discuss their outcomes in the light of the available studies related to CCA, concerning local development planning processes, based on public awareness and consensus. I would recommend the authors analytically put in evidence the advancements implied by their study as compared to the current literature, in order to make the reader aware of the value added of the submitted manuscript.
Author:
We have completely reorganized our thoughts in the discussion and conclusion section, which we combined into one critical reflection passage. In doing so, we critically confronted our results with existing literature (especially from CCA) through discussion and added citations. We moved passages that analyzed particular results into the Results section (section 3.3 Summary results) to leave the Discussion and conclusion sections for broad-overview reflections.
Reviewer:
iii. Section 5 “Conclusions.” I would recommend the authors discuss the exportability of their methodological approach to other national and international contexts different from NSB and NAB. This would imply the identification of the reasons for which the study implemented with reference to the two Alaskan boroughs is likely to be helpful in addressing similar issues related to other locations.
Author:
Please see above. Similarly, links with other geographic contexts are now discussed.
Reviewer:
iv. The English is good. However, I would recommend the authors be careful to typos here and there. For example: “The written results from this question helped to established a cultural domain of risk, an important step to CCA, to ensure culturally relevant question items later on.” (lines 261-263).
Author:
Thank you for pointing to this typo, it has been corrected. We combed the text looking for other typos and hope that we eliminated all.
Round 2
Reviewer 2 Report
All the comments and recommendations raised in the first place are appropriately addressed in the revised version of the manuscript. As a consequence, I would suggest Sustainability accept the study in its current form.